# The Sheep as a Large Animal Model for the Investigation and Treatment of Human Disorders

**DOI:** 10.3390/biology11091251

**Published:** 2022-08-23

**Authors:** Ashik Banstola, John N. J. Reynolds

**Affiliations:** 1Department of Anatomy, School of Biomedical Sciences, University of Otago, Dunedin 9054, New Zealand; 2Brain Health Research Centre, University of Otago, Dunedin 9054, New Zealand

**Keywords:** biomedical, human diseases, large animal model, sheep, translation

## Abstract

**Simple Summary:**

We review the value of large animal models for improving the translation of biomedical research for human application, focusing primarily on sheep.

**Abstract:**

An essential aim of biomedical research is to translate basic science information obtained from preclinical research using small and large animal models into clinical practice for the benefit of humans. Research on rodent models has enhanced our understanding of complex pathophysiology, thus providing potential translational pathways. However, the success of translating drugs from pre-clinical to clinical therapy has been poor, partly due to the choice of experimental model. The sheep model, in particular, is being increasingly applied to the field of biomedical research and is arguably one of the most influential models of human organ systems. It has provided essential tools and insights into cardiovascular disorder, orthopaedic examination, reproduction, gene therapy, and new insights into neurodegenerative research. Unlike the widely adopted rodent model, the use of the sheep model has an advantage over improving neuroscientific translation, in particular due to its large body size, gyrencephalic brain, long lifespan, more extended gestation period, and similarities in neuroanatomical structures to humans. This review aims to summarise the current status of sheep to model various human diseases and enable researchers to make informed decisions when considering sheep as a human biomedical model.

## 1. Introduction

A desired endpoint of preclinical research is the translation to clinical investigation of novel effective treatments and possible cures for diseases [1,2]. Biomedical research progresses towards such translation by providing knowledge of the efficacy and safety of any novel pharmaceutical molecules or medical devices [3,4]. Before human trials, new therapeutic agents need to be tested adequately for parameters such as dose range, route, and toxicity using a range of models, from the biological, using whole animals or tissues from same, to the non-biological, such as mathematical or computational [5,6]. A diverse range of experimental animal models allows a more robust understanding of disease mechanisms, capturing every aspect of disease manifestation at the genotypic and phenotypic levels [7]. Therefore, validated animal models must bridge the translational gap from the bench to the clinic and mimic the human pathological condition [1,3,4].

Animal models are numerous and stem from many different species. Small animal models, especially rodents, have been preferred in phenotype and pathology due to their low cost, ease of genetic manipulation, and rapid reproductive cycles [6]. Large animal models may not have these advantages but can be scientifically justifiable by resembling more closely human disorders than achievable in small animals [1,8]. Here, we define large experimental animal models as any non-rodent mammalian animal species currently used as an animal model for translational research, often modelling human diseases, with a focus in this review on sheep, pigs, and horses as alternatives to less readily obtainable non-human primates (NHP).

Sheep (*Ovies aries*) is a ruminant mammal, quadrupedal, livestock animal with a long history of domestication and civilization [9,10]. Over the years, Australia, New Zealand, and the USA, among other countries, have contributed significantly to the use of sheep in biomedical research [10,11], and this is rapidly growing. The use of sheep during each decade peaked in the 1970s and has been consistent ever since (see Figure 1A). Sheep have been used mainly for nervous system disorders, followed by cardiovascular and respiratory tract diseases (see Figure 1B). There have been recent reviews highlighting naturally occurring and experimentally induced sheep models focusing on neurological [12] and rare genetic disorders [7]. The focus of the present review is twofold. First, to critically review why large animal models are necessary, highlighting the factors to consider in selecting an animal model. The second is to review the current status of sheep as an experimental model to investigate various types of human organ diseases. We hope this review enables researchers to make informed decisions when considering sheep as a human biomedical model.

## 2. Criteria for Choosing a Suitable Experimental Animal Model for Translational Research

According to the National Institutes of Health, at least 80–90% of novel drugs are ineffective in clinical trials [2,3]. One of the factors contributing to this disappointing statistic is the typical selection of a small animal model (e.g., mice, rats, and guinea pigs) to study, which may not be ideal for translating therapies into clinical research [1,3,13]. In general, larger animal models can range from medium (rabbits, cats, and dogs) to large (pig, horses, sheep, and NHP) [14]. It is almost impossible to build a single animal model that perfectly reproduces the symptoms of human disorders [4,15,16,17]. Each model has its advantages and disadvantages. Thus, using several different animal models provides greater insight into other aspects of disease manifestation.

There are no stringent rules regarding the selection of an appropriate animal model. However, regulatory authorities such as the Food and Drug Administration (FDA), European Medicines Agency, and other similar international authorities require an animal model to mimic the aetiology, pathophysiology, symptomatology, and response to therapeutic interventions in humans [4,18,19]. Additionally, before human trials, these agencies expect the testing of drugs and medical devices and the demonstration of biological efficacy in at least two different mammalian animal species: one small rodent species and one other large, nonrodent species [3,4,19,20].

Before selecting one animal model over another, one must consider each candidate species’ comparative anatomical, physiological, and pathological features and how these aspects of biology resemble those in humans. Generally, several parameters are taken into account when selecting an animal model, as summarised in Table 1.

By way of example, we will briefly consider these parameters in reference to some common medium to large animal models. First, pigs are widely accepted models of digestive disease [24,25]. However, there are anatomical differences, such as the lack of an appendix and a large and delineated caecum, that need to be considered when using the pig as a model for digestive research [8,25]. In addition, their growth is comparatively fast once they are fully mature [8,25]. Similarly, pigs are used to study atherosclerosis because of the spontaneous development of lesions on the cardiac valve and vasculature [16,26,27]. However, the pig model may not be suitable for research in coronary heart disease [16,26]. Since they exhibit a significant increase in body mass over time, they are particularly prone to malignant ventricular dysrhythmias [26,28]. In addition, the time points of coronary occlusions are unpredictable [26]. Pigs are also used to evaluate the biomaterial for bone regeneration and dental implants [14,19,29] and proteomics [30]. However, pigs are susceptible to disease induction processes such as aneuploidy, polycystic ovarian syndrome, and severe combined immunodeficiency disease [24,31,32,33]. The pig is increasingly used in neuroscience; neurological interventions have been approved by regulatory authorities for clinical testing because of the similarities of the brain with humans [34,35]. Since anatomical considerations are vital for neurosurgical interventions, use of the landrace pig is more challenging because of the thick skull, which varies dramatically in size and shape [36]. However, a miniature pig might be preferable for translational research using clinical MRI protocols due to smaller skull size [37].

Second, horses are considered a highly suitable large animal model to study disease progression in osteoarthritis [38,39,40,41]. They are almost exclusively used to investigate potential therapeutic agents for the treatment of osteoarthritis, because they have a naturally occurring disease similar to humans [39,42], exhibiting a spontaneous progression of osteoarthritis without experimental induction or ectopic calcification [26,38,42]. Similarly, the incidence of melanoma, which occurs naturally in 80% of adult grey horses (12 years and older), has rendered them a suitable model for studying gene therapy approaches in melanoma treatment [40,41,43]. However, they are expensive to acquire and maintain and require a highly specialised centre for husbandry. Additionally, companion animals are a less desirable candidate for translational research [15,17].

Considering now medium-sized animals, dogs and cats are clinically more similar to humans than small animals with respect to brain and body size, prolonged lifespan, inherited and naturally occurring hereditary disorders, and surgical intervention [44,45,46,47]. As companion animals, they have the added advantage of living in similar conditions to humans [17,47]. Naturally occurring tumours in cats and dogs make them suitable models for human cancer research [47,48]. Recent studies have shown similar genome sequencing between canines and humans compared to mouse counterparts in reference to the genetics of cancers [47,48]. However, their status as companion animals, emotional impact, ethical and legal aspects, and public opinion limit their current application in translational research [17,47,49]. Other medium-sized animal models are possible but generally do not resemble the anatomical features of humans as closely as dogs and cats. The readers are referred to other reviews for information relative to medium-sized animal models.

Non-human primates such as rhesus, cynomolgus macaque, and marmoset monkeys are considered ideal models for conducting safety and efficacy evaluations of biopharmaceuticals and medical devices [50]. In cognitive and behavioural domains, NHPs show “humanlike” capabilities and self-awareness, thus contributing to research in psychology and neuroscience [51]. These animals are not easily replaced with smaller species because of their evolutionary, phylogenetic, anatomical, and physiological similarities to humans [52]. In addition, these animals form strong social bonds with researchers, display emotions and empathy, and behave with sympathy. Nevertheless, the challenges of navigating numerous ethical regulations, issues of the validity of using study groups of small sizes in biomedical research, public scrutiny, and higher cost and availability, have limited their use in research and development [50,51,52].

Table 2 compares important physiological variables of rodents, sheep, and humans. It is clear from the similarities evident in Table 2 that initial studies in rodents are helpful to provide proof of concept. However, the apparent differences when comparing small to large animals suggest it is essential to consider selecting clinically relevant large animal models to improve the likelihood of the successful clinical translation of potential novel approaches [53]. On balance, a large agricultural animal model may be more globally acceptable than a medium or large companion animal when considering the factors considered in this section.

## 3. The Use of Sheep to Model Human Diseases in Biomedical Research

Due to their long history of domestication, sheep have been used initially in the fields of dairy, animal reproduction, veterinary sciences, and more recently, in fundamental and biomedical studies [66]. In New Zealand, which has a vast population of sheep (approximately 30 million), these animals are readily available, less expensive to purchase and manage, and are cost-effective large animal models [12,66,67]. Additionally, New Zealand is home to approximately 21 different sheep breeds [68].

Sheep share many anatomical and physiological similarities with humans [66,69]. They are relatively long-living mammals, having lifespans similar to macaque monkeys, ranging between 10 and 20 years [12,69]. Besides the more comparable brain size and body weight between sheep and humans, sheep have some advantages and disadvantages as an experimental model for translational research, summarised in Table 3. 

Experimental investigations on sheep have produced a comprehensive understanding of anatomy and physiology that has built much of the necessary foundation for testing sheep as a model for human diseases [58,69]. In the following sections, we will detail this (see Table 4). In addition to their use as a model physiological system, sheep are also useful for testing the efficacy and safety of formulations and technologies and for vaccine development [77].

### 3.1. Central Nervous System Research

The study of animals with brain sizes comparatively closer to humans may be a key steppingstone for effective neurological therapeutic translation [1,3]. Large animal models can be used to overcome the failure of current preclinical research limitations through the refinement of drug development techniques, dosage regimen, pharmacokinetic, pharmacodynamic, detailed surgical procedures, anaesthetic and monitoring protocols, extensive behavioural studies, and safety and efficacy [3,4,110].

The key aspects that make sheep an acceptable animal model for neurological studies are anatomical similarities to humans, such as the distribution of cerebral white matter [53,71], a high degree of gyrencephalic cerebral cortexes, thick meninges, and the presence of highly distinct sulci and gyri [57,58,127], which contrasts with the lissencephalic cortex seen in rodents [53,57,72]. In addition, the cerebral cortices of sheep contain four lobes defined by external landmarks, similar to those of humans [128]. Furthermore, the structure of the dorsal striatum (with two separate sections: caudate nucleus and putamen) in sheep is similar to humans, in contrast to rodents, where it is not demarcated [58,129]. Moreover, sheep have a relatively round skull that provides comparable structural proximity to the human head, unlike pigs with a flat and thick skull [73]. Therefore, the brains of large animal models and sheep, in particular, may have distinct anatomical advantages over small brains for translational research.

Worldwide, noncommunicable diseases, including neurological disorders, are the leading cause of death [130], overtaking infectious diseases [131]. The most significant contributors are Alzheimer’s and Parkinson’s diseases, traumatic brain injury, and stroke, accounting for a substantial burden of morbidity and mortality worldwide [53,131]. Thus, the world health organisation has declared neurological disorders one of the greatest threats to public health [131].

Despite advances in molecular biology and biotechnology, understanding the biology of disease and the development of treatment for neurological disorders has proven challenging. There has been significant progress in developing therapeutic strategies in rodent models of human disease, but their actual value in predicting clinical outcomes does not always translate to humans [132,133]. In addition, most neurological disorders are progressive, age-related and chronic [134,135,136,137]. Thus, an animal model of large brain volume with distinct neuroanatomical structures and a longer lifespan similar to humans will play a key role in translational neuroscientific research [12,53,69,97]. A number of neurological disorders result from damage to the cortex and basal ganglia. Therefore, a model should feature a prominent motor cortex [138,139]; clearly delineated caudate and putamen [58,66,140] and substantia nigra [66]; and gross organisation and laminar structures of subventricular, subgranular zone [90], and hippocampus [141], similar to humans to provide a more accurate representation and understanding of pathological pathways and progression that may be affected in human disease. Therefore, ovine animal models could be crucial for the development of diagnostics, treatments, and eventual cures for debilitating neurological disorders. The following section provides a brief explanation of common neurological disorders and the illustration of sheep as a possible experimental model.

Parkinson’s disease (PD) is a neurogenerative disorder characterized by loss of dopamine in the basal ganglia, leading to progressive motor signs (tremor, rigidity, bradykinesia, and imbalance) and nonmotor signs and symptoms (cognitive and behavioural problems including depression, apathy, sleep disturbances, gastrointestinal, and olfactory dysfunction) [135,142]. Parkinsonian-like signs can be induced in sheep by the systemic administration of 1-methyl-4-phenyl-1,2,3,6-tetrahydropyridine (MPTP), which is transformed through enzymatic action into the neurotoxin MPP^+^ and destroys the dopamine neurons in the substantia nigra [74,96]. Hammock’s sheep study was probably the first bilateral MPTP sheep model of PD following the infusion of MPTP and MPP^+^ through jugular venous infusion. They demonstrated that MPTP caused parkinsonian behaviour in a sheep similar to that observed in NHPs [96]. A later study aimed to develop a unilateral MPTP model for PD in sheep using acute (over 30 min) and chronic (over one week) slow infusion of MPTP via a unilateral intracarotid artery, combined with occipital artery occlusion [74]. The dopamine was, however, still significant on both sides of the brain. They found that sheep displayed parkinsonian behaviour, responsive to apomorphine and amphetamine challenge, accompanied by an extensively reduced number of nigral dopamine neurons and severe depletion of caudate dopamine. However, clinical signs developed rapidly and were often severe because of the inability to prevent bilateral lesioning, raising welfare concerns. Additionally, safety considerations are required with MPTP because of the toxic nature of chemicals to both investigator and animal caretaker [143]. To our knowledge, there has not been a single peer-reviewed published study using 6-OHDA to create a sheep model of PD. Lentz and colleagues (2015) developed a novel sheep model of STN-DBS to quantify the stimulation evoked motor behaviour and found low stimulation frequency (5–30 Hz) evoked motor behaviour similar to human tremor or dyskinesia [144]. Sheep, therefore, offer the opportunity for a suitable model of human PD because of their large brain and body weight; however, the slow degeneration of dopamine neurons has yet to be emulated.

Huntington’s disease (HD), is an inherited disorder characterised by progressive cell loss within basal ganglia and cerebral cortex, leading to a hyperkinetic disorder and cognitive decline towards dementia, which is eventually fatal [58,75,145,146]. A number of different animal models have been generated, with mice being the most widely studied model. However, the mouse brain is significantly anatomically distinct to humans, and their lifespan (2.5–3.5 y maximum) is relatively short [75] to study the progressive disease. Therefore, sheep have been recently selected for development of a large animal transgenic HD model because of the developed cortex and basal ganglia [58,75]. Sheep were injected with OVT_73_ (*Ovine Transgenic* with 73 CAG repeats) that causes juvenile-onset HD and carries a CAG trinucleotide repeat seen in human disease, unlike the much longer CAG repeats used in rodent models to produce transgenic founders. Offspring were analysed at different ages (1–7 months) to reveal the robust full-length human HTT protein [75]. The immunohistochemical analysis of the caudate nucleus and putamen also showed decreased expression of markers for medium spiny neurons at later ages. Pfister and colleagues (2018) reported the successful silencing of human mutant HTT (mHTT) protein in a sheep brain by directly delivering an AAV9 carrying an artificial miRNA to the HD sheep striatum [146]. After 1 and 6 months of post-injection, the treatment reduced mHTT mRNA in the striatum. In addition, silencing was seen in both caudate and putamen without affecting the level of endogenous HTT protein. These studies demonstrated the promising application of sheep as a large animal HD model, although further development is required to ensure that the course of cell loss is more apparent within the longevity of the animal.

Alzheimer’s disease (AD) is the most common neurodegenerative disorder, characterised by progressive cognitive decline and memory impairment [147,148,149]. Later it affects older adults’ speech, motor system, and behaviour [147]. Reid and colleagues (2017) carried out a study in aged sheep (7–14 y) to characterise AD markers (amyloid-β Aβ_1–42_/ Aβ_1–42_ ratios, total tau) for determining the suitability for using sheep as a model of AD [95]. Critical protein markers were compared between humans and sheep. The authors found that plaque levels and other features were very similar and comparable. Additionally, the neurofibrillary tangles were also present in aged sheep, making the sheep a good candidate for future use in AD drug discovery and testing. As we have discussed before, the potential to develop novel therapeutics for neurodegenerative disorder (PD and AD) has been explored using sheep model, nevertheless, no sheep models are currently in active study to our knowledge. However, HD transgenic sheep have undergone trials of gene silencing therapy using miRNA to slow or halt disease progression.

Battens disease (BD), also known as neuronal ceroid lipofuscinoses (NCLs), is a family of inherited progressive neurodegenerative disorders caused by a genetic mutation in a single gene of 13 different genes characterised by visual impairment, loss of motor and cognitive functions, and seizures, leading to premature death [150,151]. A mouse model of BD exists for nearly all of these disorders (a detailed list of these models is reviewed elsewhere) [150]. However, the neurodevelopmental differences, phenotypic presentation of the disease, particularly retinal degeneration, and the short lifespan of mice compared with humans creates a gap between mouse model and clinic, resulting in translational challenges [150,152,153]. A large animal model has played an essential role in understanding the disease’s mechanism and is extremely valuable for developing diseases hallmark where mouse models have failed [152]. Jolly and colleagues (1992) first identified the sheep model of BD in 1976 using South Hampshire sheep that closely match the juvenile human disease [154]. These sheep presented with retinal degeneration signs, lipo-pigment accumulations, epilepsy, and brain atrophy at 9–12 months. Perentos and colleagues (2015) studied neurological functions and characterised sleep in CLN5 BD-affected Borderdale using electroencephalography. The epileptiform activity and mild sleep abnormalities seen in BD-affected sheep resembled those seen in children affected with BD [97]. Another study demonstrated the long-term efficacy of gene therapy in a sheep model of CLN5 BD [99]. Pre-clinically affected BD sheep (7 months old) were injected (intracerebroventricular) with single-stranded adeno-associated virus 9 (ssAAV9) or lentivirus vectors expressing ovine CLN5. They reported conserved neurological and cognitive function (when tested in an open field), increased lifespan (one ssAAV9-treated sheep survived until 62 months old, three times longer than the average lifespan of untreated BD sheep) and delayed the onset of visual deficits (20–24 months). In addition, MRI and CT scans showed stable brain volumes and brain structures. These findings support the efficacy of CLN5 gene therapy in a sheep model of BD.

In addition to measurements of gross motor function and physiological recordings of brain activity, sheep can also be used in behavioural tasks, similar to rodents and NHPs. Thus, they have been used in cognitive function tests such as decision-making [69], face recognition [155], and measurements of emotion [156]. Sheep therefore are a significantly flexible model for measurements of neurological function, emulating human health and disease processes.

### 3.2. Cardiovascular Research

Sheep have been adopted as a pre-clinical large animal for cardiovascular scientific research [54,79,81]. Ovine hearts are close in size and weight to human hearts [54]. Sheep resemble humans in terms of mechanical properties and hemodynamic flow parameters [78,157]. Additionally, growth and remodelling of the heart valve is relatively rapid (within several months in juvenile sheep), meaning that sheep are the FDA-approved pre-clinical, gold standard animal model in translational research for tissue-engineered heart valve replacements [79]. Sheep pulmonary valve stent models show good structural and functional outcomes, avoiding stent fractures and remaining intact to the pulmonary artery wall under microscopic or radiographic examination [80]. The latter study validated the implantation of a newly modified, self-expandable “Z” stent valve into the pulmonary position in sheep. In addition, more recently a sheep model of atrioventricular block was developed for the application of novel therapies [81]. Therefore, there is an increasing use of sheep to model cardiovascular research due to the similar molecular basis of valvular anatomy, coronary venous anatomy, and cardiac contraction to humans, with some exceptions [158,159].

### 3.3. Endocrine and Reproductive Research

Sheep, due to their large body size can be used for collecting detailed and repetitive hormonal profiling, examining placental development, monitoring ovarian follicular dynamics via ultrasound, and multiple neurotransmitter measures, superior to that obtainable from rodent models [83,84,86]. The reproductive developmental trajectory of sheep follows a similar time course to humans (see Figure 2). Although gestational length differs somewhat (human: 280 days; sheep: 147 days), gestational events occur at comparable time points during gestation in both species: 18, 32, and 40% of gestation in humans and 20, 37, and 51% in sheep. Similarities between sheep and humans, including those related to pregnancy [86] and common mediators of the dysfunction at the reproductive and metabolic levels, have been reviewed elsewhere [84].

For more than 40 years, pregnant sheep have been used to investigate maternal–foetal interactions [85,86]. They are very suitable for the planning of surgery and manipulations affecting the maternal and foetal vasculature [86,160]. In addition, despite structural differences there are functional similarities between human and sheep placentae [86]. The sheep placenta makes an ideal model for the human due to similar placental vascular development, placental nutrient exchange, and collection of repetitive foetal and maternal blood samples [86]. Furthermore, the sheep model provides an opportunity to test certain drugs during the prenatal period [85]. The information from these studies has informed the development of improved medicine for reproductive health.

### 3.4. Immune System Research

The immune system features a cascade of complex molecular and cellular events involving subsets of monoclonal antibodies (Mabs) and peripheral blood mononuclear cells responsible for inducing immune responses for host defence [153,161]. It involves the interaction of lymphoid and myeloid cells migrating from lymph nodes [162]. The cell-mediated immune response pathway is enhanced in sheep and goats [161]. A portfolio of monoclonal antibodies (Mabs) was produced 37 years ago to identify lymphoid cell subsets in sheep [153]. Sheep provide an opportunity to perform lymphatic cannulation to collect cells from lymph nodes, which is not possible with small rodents. The patency of cannula can be maintained for a more extended period to study the kinetics of regional immune responses in a longitudinal study and examine the dynamic of the antigen challenge [162].

Naturally occurring viral infections such as endogenous retroviruses in sheep provide opportunities to examine further the induction of immune tolerance [163]. Chemokines such as Interleukin-8 produced by macrophages and other cell types are good examples of innate immunity associated with inflammation pathology. The gene for Interleukin-8 is present in sheep while it is absent in small rodents such as mice, providing a valuable tool in advancing clinical application in humans [162].

### 3.5. Gastrointestinal Research

Sheep are ruminants with one stomach and four chambers, therefore, overall, the gastrointestinal tract structures are fundamentally different to humans [164,165]. However, anatomically, the small intestine closely resembles that of humans [164]. Other anatomic and physiologic similarities between humans and sheep include the rectum and colon, which have several lymphoid follicles resembling Peyer’s patches [166]. The sheep rectum has gross and microscopic similarities comprised of single columnar epithelial cell layers covering mucosal and crypts surface, similar to that seen in the human [107].

### 3.6. Respiratory System Research

The size and organisation of sheep lungs are close to those of human lungs, facilitating experimental procedures in sheep that are an improvement to those that can be performed in mice [167]. The bronchioles and alveoli transition, airway branching pattern and distribution of respiratory epithelial cell types are more similar, providing opportunities for respiratory diseases researchers [167,168]. Naturally occurring lung cancers such as ovine pulmonary adenocarcinoma caused by retrovirus infection, activate common cell signalling pathways and share standard features with adenocarcinoma in humans [110,167]. In addition, the ability to bronchoscopically install bacteria such as Pseudomonas aeruginosa into the lung provides a platform for the investigation of the pathophysiology of such lung infections in humans [168].

### 3.7. Ophthalmic Research

A few studies have been carried out using the sheep eye as a model for ocular research [111,113]. The sheep model was chosen because of the dimensions of its visual anatomy [111,114]. The sheep’s eye is grossly similar in structure to the human’s eye; however, it is 30% larger [113], which has advantages, especially for surgical interventions [112], or for the sampling of large volumes of aqueous and vitreous humour, as well as corneal tissue [111]. The larger globe size (vertical diameter 30 mm; cf. human 23 mm) and resemblance to human globe anatomy enables the implantation of an intraocular electrical retinal stimulator [113]. Furthermore, sheep have been used as a corneal transplantation model. However, corneal graft rejection is histologically and macroscopically similar to human corneal graft rejection [114].

### 3.8. Musculoskeletal Research

The requirements for translational orthopaedic research are an animal model with adequate size of limbs, joints and bony segments that are similar to human dimensions [115,116,117]. The sheep appears to be an excellent larger animal model for studying the musculoskeletal system [117,169], because the distribution of mechanical loads acting across the joints during load-bearing activities resembles humans. Musculoskeletal diseases such as primary muscular disorders, spinal infections, osteomyelitis, and pathological conditions such as limb lengthening, repairs of fractures, osteoporosis, and osteoarthritis have all been studied in the sheep [117]. In particular, collagen-induced arthritis in ovine models shows similar vital features such as mononuclear cell filtration, local cytokine production, angiogenesis, and synovial histopathological changes to the human [115]. Serial sampling of synovial fluid, surgical approaches, and spinal implants use similar orthopaedic instrumentation to humans, representing other advantages of the ovine model [115,169].

A study evaluating the lumbar parts of the sheep spine showed a remarkable similarity to human spinal orthopaedic research, in terms of the suitability of the vertebrate endplates and spinal canal for studying artificial intervertebral discs and implantation of intervertebral fusion [118,170,171]. Furthermore, the thoracic and lumbar regions are appropriate anatomical sites for spinal instrumentation due to similarity in dimensions [118,171]. A study investigating the early healing of cancellous bone defects found that the medial distal femoral and proximal tibial epiphyses of aged sheep (5 years old) are a suitable large animal model for human age-related degeneration when compared to that seen in the immature sheep (18 months old), a feature that is essential for future orthopaedic research [120]. Hence, sheep models have a comparable bone healing rate to that of humans [169] making them a useful large animal model in the orthopaedics field.

### 3.9. Skin Research

Dermatology research involves the testing of various agents for safety and efficacy to be applied to the human body, and for restoring the integrity of the skin after injury using different animal models [123,172]. The surgical wound model in sheep has been applied to wound healing studies because it allows for the creation of superficial, relatively large lesions, deep wounds, burn injuries, or decubitus ulcers [18,122,123,173].

Badis and Omar [122] used a sheep wound model. They showed how the topical administration of platelet-rich plasma improved the skin healing process by promoting epithelialisation after three weeks of wounding. A similar model was used recently by Martinello and colleagues (2018) to compare secondary intention wound healing after treatment with topical allogeneic mesenchymal stem cells MSCs [123]. They reported that peripheral blood-derived MSC improved the quality of skin injury and deep lesions by proliferation, neovascularisation, and re-epithelization, compared to control group at 42 days after wounding. Iacopetti and colleagues [172] conducted an identical study to compare commercially available hyaluronic acid, Manuka honey, or Acemannan gel for secondary intention healing in an ovine model. The results from the study suggested that Manuka honey enhanced the healing process, hyaluronic acid stimulated proliferation and granulation, while Acemannan gel resulted in wound drying at 42 days.

In addition to innovative therapeutic methods such as MSCs [123], ovine models have also been used in regenerative medicine for the application of low-temperature atmospheric pressure cold plasma ionised gas [174]; and dual combination of MSCs and cold plasma [175] for treating an extensive and chronic wound by improving the regenerative skin process. In addition, highly biocompatible, native spider silk fibre used in the sheep wound models showed an innovative alternative method for wound dressing in particular burn wounds [173].

### 3.10. Renal Disease Research

Renal disease is common, and there is a lack of disease-modifying therapies available [126,176]. Due to the physiology and pathophysiology of kidneys, sheep are highly suitable models for human renal diseases [125]. The advantages of sheep to study renal diseases are multiple, including size, multipapillary kidneys of sheep, and a similarity with humans in hemodynamic and coagulation systems [125,126]. The first sheep model of repeated haemodialysis treatment after bilateral nephrectomy was established in 2018 [126,177]. Before this, only goats and dogs had been used for repeated haemodialysis after nephrectomy. In this study, the two-step bilateral nephrectomy was conducted to develop an ovine model of repeated haemodialysis treatment to measure dialysis adequacy and urea reduction ratio for each haemodialysis treatment without pharmacological intervention. In addition, a study by O’Kane and colleagues [20] demonstrated the intravenous injection of 0.5 mg/kg/day of ZnCl2 dosages protected against renal ischaemia reperfusion injury in sheep [20]. In contrast, rodent studies of remote ischaemic preconditioning and direct intermittent arterial clamping to reduce renal ischaemia reperfusion injury have not been successful in translation to humans. This, again, highlights the importance of large animal models over small laboratory animals [178,179].

## 4. Technical Advances

### 4.1. Vaccine Development and Testing

As discussed earlier, regulatory authorities require drug and vaccine candidates to undergo pre-clinical testing before going into the clinical testing phase. Sheep, among other large animals, have been valuable for vaccine development and testing [77]. One of the essential advantages of using sheep as a model for vaccine development is that sheep are a natural model for various human infections, for instance, para influenza virus [180], *E. coli* [181], Brucellosis ovis [182], and respiratory syncytial virus [180]. The outbred nature of sheep more closely mimics the human vaccination programme, where mixed populations are tested, yielding mild, moderate, or non-responders to a given vaccine. Furthermore, the neonatal period in sheep is comparable to humans, making it suitable for determining the effective vaccine dose and evaluating vaccine efficacy [77]. Moreover, pregnant sheep can be immunised with various experimental vaccine candidates to understand better the placental transfer of maternal antibodies to foetal lambs [183].

### 4.2. Therapeutic Interventions

In terms of neurological interventions, the sheep brain and skull closely resemble humans in terms of anatomical features such as a thick cancellous skull, porosity, and curvature [184,185]. This is unlike pig skulls which are flat and thick, with thickness increasing with age [186]. Investigating the effects of high-intensity, focused, non-invasive ultrasonographic brain surgery on sheep, Pernot and colleagues [185] were able to induce thermal lesions in deep structures (30–40 mm deep) by concentrating an ultrasound beam of sufficient intensity through the skull. The authors confirmed the induction of thermal lesions and noted that the approach did not affect the neurological function of the animal. Thus, sheep are feasible as an animal model for performing non-invasive and non-ionizing treatment for neurological disorders and tumours.

Similarly, sheep have been established as a novel animal model for non-invasive, ultrasound-assisted blood–brain barrier opening through the intact skull for drug delivery [184]. In addition, sheep have been used as a preferred large animal model for the development of fully implantable (placed sub or epidurally on the cortex), wireless closed-loop devices (known as Braincon) suitable for both stimulation and recordings of neural activity [187]. These observations suggest that sheep are a convenient translational tool for testing the efficacy and risk of implants for chronic human clinical applications.

## 5. Limitations and Future Directions

This review has emphasised sheep models for studying human diseases. However, some issues with using sheep as a large animal model need to be considered before planning such studies. Firstly, sheep may be readily available and cost-effective on the initial purchase but may incur a relatively high expense for maintenance for more extended studies. Although sheep spend most of their time grazing in a natural pasture, other assistance is required. For instance, animal husbandry, the construction of holding areas and operating facilities suitable for sterile operation (if not already available) need a highly skilled surgical team, and veterinary assistance before, during and after the manipulations, increasing the total experimental cost. Secondly, unlike rodents and NHPs, biochemical and molecular data are less available, and obtaining appropriate antibodies for routine immunohistochemistry can be challenging and costly. Such costs could be considerable with the application of sheep to chronic and progressive neurological research, for which they are a preferred model due to their long lifespan and larger brains. Hence, research costs need careful consideration for translational and more extended studies.

Thirdly, considering specifically neurodegenerative diseases, there are a number of anatomical challenges to highlight. While a large animal cerebral blood supply may better mimic the human condition than rodent brains, cerebral blood supply varies between all species. For instance, sheep have a rete mirabilis whereas humans do not, and how this structure may influence cerebral blood flow following traumatic brain injury has not been characterised. Hence, blood supply is another factor to take into account when considering sheep as a model for cerebrovascular diseases [76]. Further, considering the comparative anatomy of the brain itself, a detailed sheep brain atlas with full stereotaxic coordinates system is not available to facilitate functional mapping of sheep to human brain nuclei [188]. Instead, the available ovine brain atlas is limited to specific areas. This could be expanded by labelling more structures and including other breeds, in addition to the commonly used merino sheep [189]. The more intensive application of available neuroimaging techniques (MRI, fMRI, and DTI) and the subsequent production of a detailed atlas could further enhance the use of a sheep model in the translational neuro research [73].

Finally, the major limitation of work on large animal models, including sheep, is the lack of genetic models. The ovine genome has not been fully sequenced, and molecular tools are limited to study, e.g., brain injury or digestive research compared to the information available for rodents [8]. Furthermore, unlike rodents, standard behavioural testing protocols are limited in sheep models.

## 6. Conclusions

The large animal experimental model provides translational information helpful in understanding disease progression over time or the mechanism of therapy in human studies compared to rodents. Most biomedical research uses female sheep for surgical procedures and testing medical devices as they are calm and easy to handle compared to rams. In those instances, a researcher would be expected to take into account physiological fluctuations in female hormonal levels if they bear relevance to the study.

The use of sheep for human biomedical models is summarised in Figure 3, including cardiovascular surgery; atrioventricular block model; model for human pregnancy; ovine pulmonary adenocarcinoma model for human lung cancer; skin wound healing model; skeletomuscular disorder; ophthalmic model; and neurological disorders such as partial epilepsy, stroke, PD, AD, BD, and HD.

The sheep model is an ideal candidate for testing human-sized emerging therapeutic applications, such as non-invasive focused ultrasound to open the blood–brain barrier, gene therapy, replacement of tissue-engineered heart halves, and uterine transplantation. The sheep model could replace NHP models in specific areas, which is especially important given the challenges of working with primate models, such as high cost and ethical concerns in some countries. However, there are several disadvantages to using large animal models that need mindful consideration before undertaking such studies. There is no specific single animal model that can present a clear picture of several human diseases. Thus, for the success of translational research, initial studies in rodents are needed to provide the proof of concept, followed by studies in large animal non-rodent mammalian species. Interspecies differences must be recognised and considered in both research design and extrapolation of research outcomes.

Determining the appropriate experimental model requires several decisions and compromises based on the nature of the research question. It is essential to standardise the procedures used to obtain relevant and reproducible results compared with other findings. In the future, we can see more extensive use of large experimental models and acceptability by regulatory authorities as a physiologically preferable stand-alone animal model rather than simply a scale up of the small animal model. The sheep model can provide an opportunity to test new pharmacological agents and medical devices that can potentially be translated to clinical applications, for example, focused ultrasound and brain stimulation technology. There are some limitations to using sheep as biomedical models: maintenance and more expensive for extended studies, fewer available antibodies, and lack of detailed atlas compared to rodents. Despite these limitations, the use of sheep in basic science, applied technologies, and translational medicine could fill the knowledge gaps between studies in small models and success in human trials.

## Figures and Tables

**Figure 1 biology-11-01251-f001:**
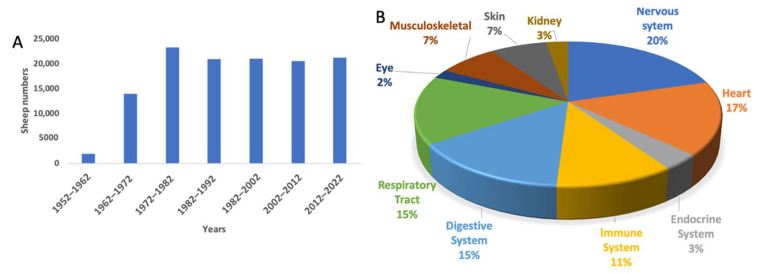
Application of sheep to biomedical research. (**A**) The number of sheep used internationally per decade. (**B**) The pathological conditions for which sheep models have been applied since 1952.

**Figure 2 biology-11-01251-f002:**
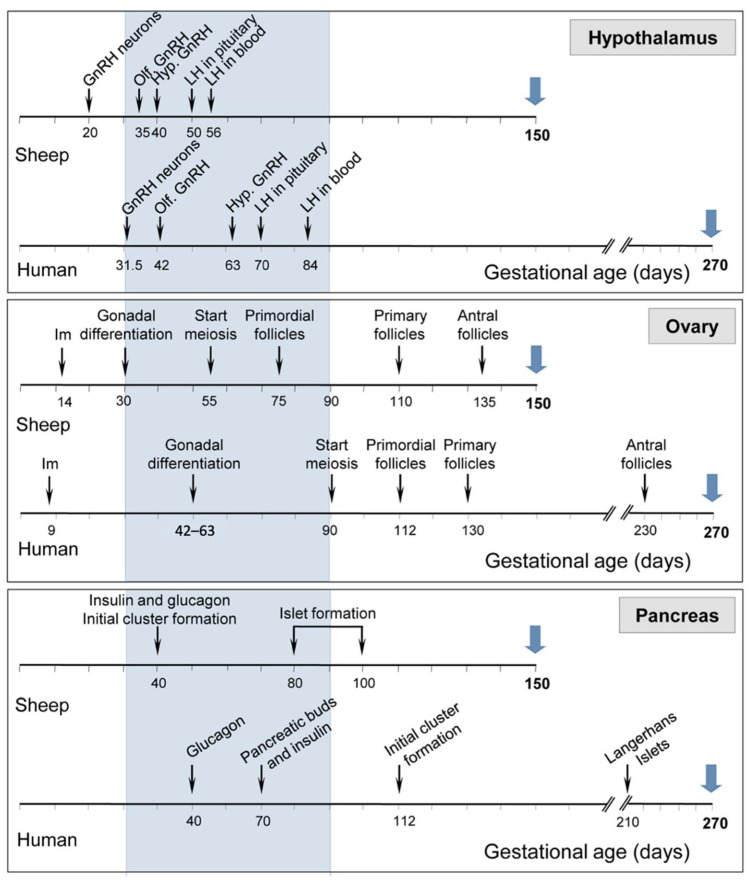
Comparison in sheep and humans of the timing of development of neural circuitry in the hypothalamus (**top** panel), follicular differentiation in the ovary (**middle** panel), and secreting cells in the pancreas (**bottom** panel). Blue arrows indicate gestational length. Hypothalamus: gonadotrophin-releasing hormone (GnRH) neurons: appearance of first GnRH immunoreactive neurons, Olf. GnRH: GnRH neurons visible in olfactory bulb, and Hyp. GnRH: appearance of GnRH neurons in the hypothalamus. Ovary: Im: implantation. Figure adapted from sheep models of polycystic ovary syndrome phenotype by Padmanabhan et al. [84] with permission from Elsevier.

**Figure 3 biology-11-01251-f003:**
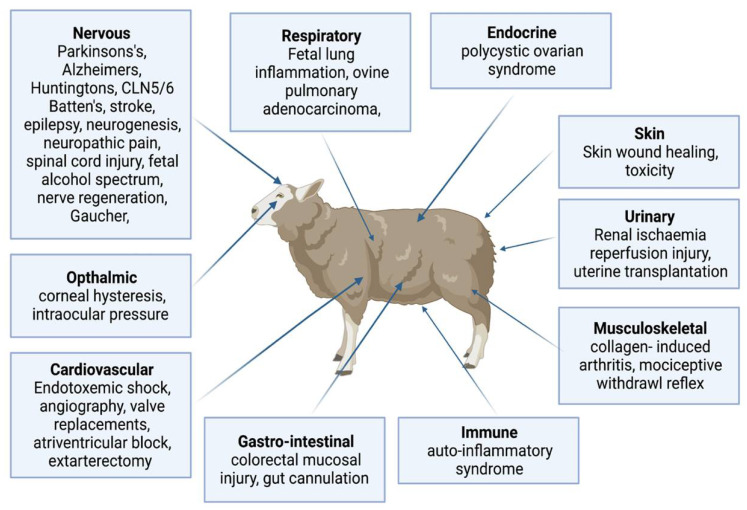
Sheep biomedical models for human physiology and diseases (Created with biorender.com accessed date 6 May 2022). The sheep is a large-experimental animal model to study normal anatomy, physiology and disease mechanism across various body systems such as the cardiovascular, respiratory, endocrine, urinary, musculoskeletal, immune, gastro-intestine, ophthalmic and neurological disorders such as partial epilepsy, stroke, Parkinson’s, Alzheimer’s, Batten, and Huntington’s.

**Table 1 biology-11-01251-t001:** Criteria for choosing the most appropriate animal model. Summarised from refs [6,21,22,23].

Criteria
Ethical regulations and public scrutiny Cost and availability
Anatomical and physiological comparability
Adaptability to experimental manipulation
Genetic similarity, where applicable
Behavioural phenotype
Resistance to infections and disease progression
Investigation of spontaneous (natural) or induced (experimental) response
Transferability of information between species

**Table 2 biology-11-01251-t002:** Comparison of common physiological variables of adult (male) rodents, sheep, and humans.

Physiological Variables	Rodents	Sheep	Human	Reference
Adult body weight (kg)	0.3–4.0	60–70	70–80	[54,55]
Lifespan (y)	2.5–3.5	10–30	70–80	[25,56,57]
Brain weight (kg)	0.013	0.13–0.14	1.3–1.4	[58]
Rectal temperature (°C)	38–39	38–39.5	36.7–37.5	[55,59,60]
Respiratory rate (breaths/min)	66–114	15–40	9–20	[55,59,61]
Respiratory dead space (mL)	0.6–1.25	100	150	[55,59]
Tidal volume (mL/kg)	0.6–1.25	4–9	7	[59,62]
Heart rate (beat/min)	330–480	50–88	50–100	[55,59,61]
Maximum heart rate (beats/min)	370–580	260–280	140–150	[59,62]
Mean arterial pressure (mm Hg)	119–122	70	70–105	[59,63]
Cardiac output (L/min)	0.12–0.13	1.5–13.2	4–8	[59,64,65]
Stroke volume (mL/beat)	0.41–0.43	74	60–100	[59,64]
Extracellular fluid volume (mL/kg)	300	246	260	[59]
Plasma volume (mL/kg)	30.8–36.7	37	43	[55,59]
Blood volume (mL/kg)	56–71	49	70	[55,59]
Haemoglobin (g/100 mL)	11–19 g/dL	9–15	14–16	[55,59]

**Table 3 biology-11-01251-t003:** The pros and cons of using sheep as an experimental animal model for translational research compared to other animal models.

Factors To Consider	Pros	Reference
Accessibility for research	Greater acceptability to animal ethics committees cf. companion animals	[26,53]
	Easily available, reasonably outbreed, less expensive to buy cf. other large species	[10,29,53,70]
	Long living mammal (10–20 years) with body size, weight and brain comparable to the human	[19,29,57,58,70,71,72,73]
	Easy management, Surgical manipulation	[14,29,70,74,75]
Body system specific advantages	Brain: higher degree of white matter and gyrencephalic structure, strong cerebellar tentorium cortical organization	[19,57,58,70,72,73]
	Excellent animal model to study HD due to HTT gene	[69]
	Long bones with dimensions suitable for the deployment of implant systems	[14,29,58]
	Excellent animal model to study osteoporosis due to long bones, early brain development	[14,29,70]
	Suitable to study the main physiological systems: cardiovascular, orthopedic, endocrine, respiratory, renal, nervous and reproductive systems, host organisms for virus infection	[10,14,29,58,70]
Others	Environmental enrichment not required as they live in their natural pasture	[58]
	Increased clinical translation and more accurate indication (as the dosage, drug distribution, and safety of potential therapies trialed)	[66]
	Ability to use clinically relevant technology to provide clinically translational measures, such as MRI, CT, and PET; ability to use clinical equipment such as anesthesia, physiological monitoring, surgical equipment	[53,76]
	Strong, can carry a backpack with transmitting devices	[58]
	Natural model for human infections such as parainfluenza, *E.Coli.*	[77]
	**Cons**	
Accessibility for research	Higher ethical considerations than small animals	[29,53]
	Higher costs for maintenance and larger facilities required to perform procedures than small animals	[10,29,53,70]
Body system-specific advantages	Longer gestation time compared with small animals; Uniparous in breeding, difficulty scaling up the number of offspring	[19,29,57,58,70,71,72,73]
	Not suitable for studying neurobiology of fine motor control and binocular eye movements	[14,29,70,74,75]
	Transgenic selection and production of transgenic strains limited	[69]
	Poor availability of physiological databases for mapping to humans including atlases	[14,20,70]
Others	Vital sign monitoring needed by a veterinary practitioner during all surgical procedures	[19,57,58,70,72,73]
	Unlike, rodents, no standard behavioral testing protocol	[14,29,58]
	Limited access to antibodies	[10,14,29,58,70]

**Table 4 biology-11-01251-t004:** Summary of the available sheep models of major physiological systems.

Body System	Disease or Syndrome Model	Strain/ Sex	Age/ Body Weight	Description/Observation	Reference
Cardiovascular	Ovine endo-toxemic shock (macrocirculation)	Sheep/F	Adult	The macro- and microvascular effects of selective and nonselective potassium channel inhibitors studied in ovine endotoxemic shock.	[78]
	Angiography of the cardiac coronary venous system	Lacaune/M	68 ± 5.3 kg	The general organization of the coronary venous circulation evaluated from clinical angiographic studies.	[54]
	Tissue-Engineered(TE) heart valves replacement	Swifter/F	1 y	Immunological markers and expression of proteins specific to sheep validated for the immunohistochemical analysis of tissue-engineered heart valve after implantation in a sheep model.	[79]
	Beating heart sheep model	Sheep	42 ± 5 kg	Structural and functional outcomes of trans-ventricular implantation of a modified stented bovine pulmonary valve assessed.	[80]
	Atrioventricular block model	Castrated, merino cross/M	1 y/ 63.1 ± 5.6 kg	Atrioventricular block by radiofrequency ablation of the His bundle and implantation of the pacemaker into the right ventricular apex developed, characterized and validated in adult sheep.	[81]
	Extarterectomy model	Sheep/M	1^1/2^ y/ 39–48 kg	Tumor removal by extarterectomy technique and its long term effect of on the vascular wall and arterial blood flow investigated in male sheep.	[82]
	Chronic heart failure	Sheep	NA	Testing and optimizing of surgical therapies for chronic heart failure.	[28]
Endocrine	Polycystic ovarian syndrome	Scottish Greyface/F	Adult	Prenatal testosterone (T)-treated female sheep showed reproductive deficits comparable to women with polycystic ovarian syndrome (PCOS).	[83,84]
	Maternal-Fetal sheep model	Dorse/F	Adult pregnant	a maternal-fetal pharmacokinetic model of propofol in pregnant ewes successfully developed.	[85]
	Pregnancy model	sheep	Adult pregnant	Placental development, oxygen and nutrient transfer between maternal-fetal interaction are similar to human pregnancy.	[86]
Immune	Autoimmune/autoinflammatory syndrome induced by adjuvants (ASIA syndrome)	Spanish castrated lamb/M	3 months	Repetitive inoculation of aluminium-containing adjuvants through vaccination showed acute and chronic neurological episode resulting in ASIA syndrome that can be used to model similar disease affecting both human and animals.	[87]
Nervous	Peripheral nerve regeneration	Sheep	NA	Similar size and regeneration behavior of nerves supports the use of sheep as a model for studying peripheral nerve regeneration following nerve injury.	[88]
	Neurogenesis	Sheep	NA	Distinctive feature of hypothalamic, olfactory and hippocampal neurogenesis in adult sheep and its contribution to reproduction, odour processing and maternal behavioral revealed.	[89]
	Neurogenesis	Romney/ Suffolk	3–6 y/ 53.1–59.8	Cell proliferation in the subventricular and subgranular zone of adult sheep is comparable with human and has the same distinct layers.	[90]
	Transient stroke	Merino/ M and F	18–36 months/ 65 ± 7 kg	A survival model of sheep transient middle cerebral artery occlusion and the temporal profile of intracranial pressure change following transient stroke developed in sheep.	[53,76]
	Occlusion (permanent middle cerebral artery) stroke model	Outbreed adult hornless merino /M	Adult/ 42–65 kg	The permanent middle cerebral artery (MCA) occlusion results into cerebral ischemia and produces reproducible neurologic dysfunctions and can be modified by altering the occlusions in MCA.	[91]
	Acute proximal middle cerebral artery ischemic stroke	Merino/M and F	18–24 months/50.1 ± 5.8 kg	A surgical model of permanent and transient MCA stroke in the sheep developed.	[92]
	Spina bifida	Lamb	NA	The congenital anomaly of CNS (spina bifida phenotype) with and without myelotomy comprehensively and reliably characterised in fetal lamb.	[93]
	Axonal injury	Merino/F	2 y	The physiological and pathological changes resulting from traumatic injury using immunostaining as a marker of early axonal injury developed to established a head impact model of axonal injury in sheep.	[94]
	Transgenic Huntington’s disease	Sheep/F	1 and 7 months	Six transgenic founder sheep generated, expressing full length human HTT with a poly- glutamine region of 73 residues.	[75]
	Alzheimer’s disease	Sheep	8–14 y	The processing of amyloid protein, total tau and neurofilament markers in the aged sheep comparable to those found in sheep.	[95]
	Parkinson’s diseases	Columbia-Suffolk cross bred/F	1–3 y/ 40–70 kg	Infusion of MPTP and MPP^+^ through jugular cannula produces parkinsonian-like behavior in sheep.	[96]
	Parkinson’s diseases	Rambouilette, ranch bred/F	1–3 y/ 40–55 kg	Unilateral acute (over 30 min) and chronic (over 1 week) intracarotid injection of MPTP (0.4–5.0 mg/kg) via slow continuous infusion produces Parkinsonian-like behavior in sheep.	[74]
	Sleep and neurological dysfunction	Borderdale	14.3 ± 0.5 months	Electroencephalography study performed in CLN5 batten disease-affected sheep to characterize the sleep and neurological dysfunction.	[97]
	Motor neuron syndrome	Border Leicester dominant cross/	18 months/ 34–45 kg	Molybdenum deprivation, purine ingestion and an astrocyte-associated motor neurone syndrome produced 18–27 months later in sheep.	[98]
	CLN5 Batten disease	Borderdale	2 to 3 months	Efficacy of CLN5 gene therapy on the CNS monitored in a sheep model of CLN5 batten disease.	[99]
	Focal epilepsy	Merino/F	Adult	Focal epilepsy in sheep generated with injection of penicillin into the right prefrontal cortex and studied with the use of fMRI and iEEG.	[100]
	Neuropathic pain model	Polypay sheep/F	75 kg	A neuropathic pain model established by tight ligation and axotomy of the common peroneal nerve and analgesic effect of morphine studied in sheep.	[101]
	Ovine ceroid lipofuscinosis (CLN6)	South Hampshire lamb/M and F	9–12 month	Linkage between ovine ceroid lipofuscinoses (CLN6) and microsatellite markers OAR 7 q13–15 established.	[102]
	Spinal cord injury	Suffolk	2 y	A model of spinal cord injury established by hemi sectioning of the spinal cord (left side) and injury quantified by a gait analysis of pre an post injury in ovine.	[103]
	Fetal Alcohol Spectrum Disorder	Sheep/F	NA	Pregnant sheep were exposed to binge alcohol consumption for a three-trimester period and plasma MiRNA profile was assessed from pregnant and from newborn.	[104]
	Acute neuronopathic Gaucher disease	Lamb	2 h after birth	Acute neuronopathic Gaucher disease model developed in lamb by mutation in the β-glucocerebrosidase gene C381Y, which is equivalent to human C342Y.	[105]
	Non-accidental head injury	Lamb	5–9 days old/ 5–12 kg	Axonal injury, neuronal reaction, and albumin extravasation examined in the hemispheric white matter, brainstem and at the intracranio cervical junction after manual shaking of head in lamb.	[106]
Gastro-intestinal	Colorectal Mucosal Injury	Yearling virginal/F	NA	Due to similar gross and microscopic between sheep and human rectum Optical coherence tomography imaging and colonoscopy used to visualize morphological abnormalities and scoring of microbicide-induced injury in sheep model.	[107]
	Intestinal loop model (gut cannulation)	Canadian Arcott/F	8–10 months	Surgical method of catherization of intestinal loops without affecting health or loop function developed to elucidate the host response to various treatments within the small intestine of ruminants.	[108]
Respiratory	Fetal lung inflammation	Merino/F	NA	Acute systemic inflammatory response of the 10 mg *e.coli* LPS into amniotic fluid showed three major fetal surfaces exposed to inflammatory mediators in pregnancy (the lung, gastro-intestinal tract and skin/amnion).	[109]
	Ovine pulmonary adenocarcinoma	Sheep/F	39–65 kg	Sheep model of naturally occurring lung cancer, ovine pulmonary adenocarcinoma caused by jaagsiekte sheep retrovirus has similar histological characteristics of human lung adenocarcinomas.	[110]
Ocular	Ophthalmic model	Coopworth × Texel/M	10–12 months/ 50–60 kg	The anatomical and clinical characteristics of sheep such as ocular response, central corneal thickness, topographic maps, intraocular pressure, corneal hysteresis, and corneal resistance factor confirmed the suitability of sheep as a model for ophthalmic experiments.	[111]
	Opthalmic surgery training	Afshar	1 y	Sheep eye for ophthalmic surgery training in skills laboratory found similar ultrasonic graphic and physical biometric description.	[112]
	Vision prosthesis model		NA	Development and surgical implantation of a vision prosthesis model into the ovine eye.	[113]
	Corneal transplantation	Merino	Adult/44–68 kg	Corneal graft rejection in the sheep is macroscopically and histologically similar to human corneal graft rejection.	[114]
Musculoskeletal	Collagen-induced arthritis	Merino (F)	2 y	Bovine collagen type II injected into the hock joint and the histopathological scoring system was developed to established a collagen induced arthritis model in sheep.	[115]
	Nociceptive withdrawal reflex	Swiss alpine	2–3 y/ 63.1 ± 6.1 kg	Nociceptive withdrawal reflex induced by electrical stimulation in the thoracic and pelvic limb used as a tool to evaluate nociception in conscious non-medicated sheep.	[116]
	Osteoporosis	Corriedale/F	2 y	Osteoporosis induced by hypothalamic-pituitary disconnection in sheep model to determine sustainability of bone loss and its biomechanical relevance.	[117]
	Spinal orthopedic model	Merino/F	2 y/62.5 ± 5.3 kg	CT scan from sheep L1 to L6 showed similar vertebral endplates and spinal canal to humans supporting sheep as a model for human orthopedic spinal research.	[118]
	Hypophosphatasia	Sheep	NA	Using CRISPR/Cas9,a single point mutation in the tissue nonspecific alkaline phosphatase (TNSALP) gene (*ALPL*) made to generated hypophosphatasia- rare human bone disease.	[119]
	Cancellous bone healing	Sheep	18 months and 5 y	A sheep model for cancellous bone healing surgically created to assess early healing and biological changes of the medial distal femoral and proximal tibial epipheses bilaterally.	[120]
	Artificial bone substitute	Mecklenburg	63 kg	Artificial bone substitute (Nanobone) used in bone defect model of ovine tibial metaphysis appears to be highly potent bone substitute with osteoconductive properties.	[121]
Skin	Skin wound healing model	Sheep/M	6 months/ 20–25 kg	Topical administration of platelet-rich plasma improved skin wound healing process in sheep making a good model for regenerative medicine research.	[122]
	Second intention healing model	Bergamasca/F	NA	Peripheral blood-derived MSCs improve the quality of wound healing by accelerating granulation, reepithelization and neovascularization both for superficial injuries and deep lesions.	[123]
Urinary	Renal ischaemia–reperfusion injury	Sheep/F	35–40 kg	The reno-protective effects of zinc preconditioning in renal ischaemia reperfusion injury was assessed in a sheep model and found to be beneficial for human applications.	[20]
	Uterine transplantation	Biparous (Prealpes and Romane)/F	35–70 months/ 57–91 kg	Uterine transplantation method set-up using end to end anastomosis of external iliac vessels in sheep and complications were highlighted.	[124]
	Renal function	Segurena/M and F	2–6 y/50 kg	Simplified Iohexol-based method was developed to measure renal function using glomerular filtration rate as a model of renal diseases in sheep.	[125]
	Haemodialysis treatment	Merino/M	16–18 months	The two-step bilateral nephrectomy was conducted to develop ovine model of haemodialysis treatment to measure dialysis adequacy and urea reduction ratio for each haemodialysis treatment.	[126]

## Data Availability

Not applicable.

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
