# Peer review of "The Sheep as a Large Animal Model for the Investigation and Treatment of Human Disorders"

_biology, 2022, doi:10.3390/biology11091251_

Round 1
Reviewer 1 Report
The review of Banstola and Reynolds sums up the use of sheep models in different human diseases. It is a very important topic since large animal models are important for translational testing before expensive clinical trials. The summarizing tables are very helpful for choosing the model for a certain disease. The article is well organized and written.
Some minor points.
In section 2 other large and medium animal models are discussed. There is quite a big paragraph about horses nevertheless the non-human primate models, models considered closest to humans, are not mentioned. Also, minipigs are not mentioned.
In line 108, the sentence about reliable reagents for pigs is cited with citation form 2014, which is very old, and it is not that much true any more since the sequence can be confirmed.
Also, the feature of flat and tick skull of pigs (line 107-108, 185, and 471) is true, but it is not a big limitation for neurosurgeries and MRI. It is cited by an article about MRI in sheep, there is no citation about pig (or minipig) MRI. Even though neurological interventions tested on pigs were approved by regulatory authorities for clinical testing (HD uniQure trial, Evers et al. 2018, Valles et al. 20210
Reviewer 2 Report
In this review, the authors summarised the current status of sheep to model various human
diseases and enable researchers to make informed decisions when considering sheep as a human biomedical model.
This manuscript is interesting and special; unfortunately, this manuscript needs substantial improvements and corrections before publishing may be possible.
General points:
Please add a list of abbreviations before References section to your manuscript.
Special points:
Important, this manuscript should be substantially improved, i. e., by substantial references in the field.
Introduction
Lines 32-42: please add more references at the end of each these sentences.
Lines 43-45: please add multiple references at the end of this sentence.
Lines 52-58: please add more references at the end of each these sentences.
Main part:
Lines 70-86: please add more references at the end of each these sentences.
Lines 94-129: please add more references at the end of each these sentences.
Lines 144-157: please add multiple references at the end of each these sentences.
Lines 170-180: please add more references at the end of each these sentences.
Lines 188-193: please add more references at the end of each these sentences.
Lines 198-201: please add multiple references at the end of each these sentences.
Lines 211-220: please add multiple references at the end of each these sentences.
Lines 211-233: please add to your review:
- What about the 6-OHDA model in sheep?
- Which exactly MPTP model for Parkinson’s disease was used: unilateral or bilateral?
Lines 211-263: please add to your review:
- What about the using of sheep model for development of new therapeutics for Parkinson’s disease, Alzheimer’s disease and Huntington’s disease?
Lines 234-236: please add multiple references at the end of each these sentences.
Lines 255-256: please add multiple references at the end of this sentence.
Lines 269-272: please add multiple references at the end of this sentence.
Lines 297-298: please add multiple references at the end of this sentence.
Lines 313-316: please add multiple references at the end of this sentence.
Lines 323-326: please add multiple references at the end of each these sentences.
Lines 341-344: please add multiple references at the end of each these sentences.
Lines 359-360: please add multiple references at the end of this sentence.
Lines 378-379: please add multiple references at the end of each these sentences.
Lines 389-390: please add multiple references at the end of this sentence.
Lines 401-404: please add multiple references at the end of each these sentences.
Lines 413-415: please add multiple references at the end of this sentence.
Lines 438-440: please add multiple references at the end of each these sentences.
Lines 443-444: please add multiple references at the end of this sentence.
Conclusion
Lines 525-526: you said: Most biomedical research uses female sheep, mainly for
surgical procedures and testing medical devices.
What about the hormones influence on the results? Please add to your review.
Figure 3: please add the Legend with the appropriate description to your Figure 3.
Round 2
Reviewer 2 Report
Thank you for all corrections.